# Climate Change, Addiction, and Spiritual Liberation

**Margaret Bullitt-Jonas** [1,2,3]

1   Missioner for Creation Care, Episcopal Diocese of Western Massachusetts, Springfield, MA 01103, USA;
    margaretbj@aol.com
2   Missioner for Creation Care, Southern New England Conference, United Church of Christ, Framingham,
    MA 01702, USA
3   Creation Care Advisor, Episcopal Diocese of Massachusetts, Boston, MA 02111, USA

**Abstract:** Climate scientists have sounded the alarm: The only way to preserve a planet that is generally habitable for human beings is to carry out a transformation of society at a rate and scale that are historically unprecedented. Can we do this? Will we do this? Drawing on her long-term recovery from addiction and on her decades of ministry as a climate activist, the author reflects on how understanding the dynamics of addiction and recovery might inform our efforts to protect the web of life and to bear witness to the liberating God of love who makes all things new.

**Keywords:** addiction; recovery; Twelve-Step Program; climate change

## 1. An Addict's World

The addict looks away. The addict sees but does not see. She does not want to see. There is nothing to see here. Change the subject.

The addict is empty. She does not have enough. She must be filled. She must be filled right now.

The addict carries out repetitive, compulsive rituals that disconnect her from self, others, Earth, and the sacred.

The addict functions like a machine. She repeats the same behavior over and over, despite its harmful consequences to herself and perhaps to others, too.

The addict is ruthless. She dominates, forces, and exploits. The addict treats everything, including herself, as an It.

The addict is cut off from her body. Who cares what the body wants? She ignores and overrides the body, its wisdom and needs.

The addict is cut off from the rest of the natural world.

The addict lies to herself and she lies to others. (*There is no problem here. Do you see a problem? I do not see a problem*).

The addict is numb. She does not feel.

The addict is self-centered, isolated, and alone.

The addict is used to this. This is normal. This is the way things are. Nothing will ever change.

The addict is powerless. She is trapped. She cannot stop herself. She intends to change, she plans to change, she promises to change, she tries to change. She does not change.

The addict hates herself.

Her life is unmanageable.

## 2. A Story of Recovery

Writing these words, I conjure up my state of mind forty years ago, when I was gripped by an eating disorder. As a teenager and young adult, I ate compulsively. To compensate for the binges, which I carried out in secret, I ran endless miles, tried every diet under the sun, and fasted for days on end. I made endless vows—this time I would not eat more than I needed; this time I would overcome my cravings—but my vows, however ardently expressed, had no power to set me free. Inevitably, I went back to the box of donuts, or the jar of peanut butter devoured hastily and with the shades drawn, lest anyone see me, lest I see myself.

My drug was food. As any addict knows, addiction distorts and numbs our awareness of the body. In those years of compulsive overeating, I paid little attention to my body's rhythms or needs. Feelings did not matter. So what if I was sad or lonely? So what if I was angry, excited or bored? Whatever I felt, I swallowed it down with food and set out for another grueling run. Was it night-time and was my body eager for sleep? I did not care. I would stay up late, make a tour of the all-night supermarket, and eat until my stomach ached. Was I disappointed and needing to cry, or angry and needing to be heard? Quick—I would pave over those feelings and force some cheese or chocolate down my throat. Was my body aching from the abuse I dished out? Too bad. After a bout of bingeing, I would get up the next morning and go out for a seven-mile run, maybe start another fast or launch another stringent diet. Pummel and punish the body—that was my motto. Clear-cut the forest and move on.

Like every addict who has lost control, I could not stop what I was doing, and I saw no way out. At last, through the grace of God, at the age of thirty, I found a path to recovery. Now almost seventy, I sing the familiar words of the hymn "Amazing Grace"—*I once was lost and now am found*—and look back with gratitude to 13 April 1982, the day I walked into a Twelve-Step meeting and held up the white flag of surrender: *Help. I give up. My life is unmanageable*. I could not fight the battle any longer, for it was a battle I always lost. I needed help beyond myself. I needed a Higher Power. I had to make peace with my body or die (Bullitt-Jonas 1998).

That day was the turning-point of my life, the beginning of a journey to wholeness. One day at a time, I began practicing the Twelve-Step Program of Overeaters Anonymous and dug into the physical, emotional, and spiritual work of reconciling with my body, myself, and the important people in my life. I began to take responsibility for the first bit of nature entrusted to my care—my body. Day by day I began to honor its limits and listen to its needs. I met regularly with a psychotherapist and began to untangle my inner knots. Additionally, I embarked on a spiritual search. Impelled by an intense desire to know what was real, what was lasting, trustworthy, and true, I ventured back into the church I had long ago abandoned and sat in the shadowed back pew so that I could listen from afar. I longed to know who God was, and how to meet God in my own experience. I began to study and practice meditation and prayer.

My mind, it turned out, was as jumpy as water on a hot skillet. I was surprised by the inner racket: worries, memories, regrets, and plans. Arguments, scraps of music, commercial jingles. How could I love God, my neighbor, or myself if I was perpetually distracted? I learned to bring awareness to the breath and to return to the present moment, disciplining my attention so that I could perceive more accurately what was here. As my mind settled down, strong feelings surged through me. Shame, sorrow, anger, yearning—for years, they had been tamped down in my long bout with addiction, but now, here they were, roaring back to life. I sat with the feelings and breathed, learning to give them space and let them be. The feelings ebbed and flowed. They always passed. No one died. In fact, the more I allowed them to come and go, the more spacious I felt, and the more truly alive.

Love kept showing up. When I welcomed everything into awareness, clinging to nothing and pushing nothing away, an unexpected tenderness would eventually rise up from within and gather me up like a child. I went off for a ten-day silent retreat at a

meditation center in western Massachusetts. I followed the drill: You sit. You walk. You sit. You walk. That is it. You do nothing but bring awareness to the present moment.

One day I left the retreat house for a walk in the woods. I paid attention to sensations as they came, the feel of my foot on the ground, the sound of birds, the sight of birches, hemlock, and pine. My thoughts lay still. I was nothing but eyes and ears, the weight of each foot, the breath in my nostrils. At one point I stopped walking, overwhelmed by the sense that the whole world was inside me. I was carrying the round blue planet inside my chest. My heart held the world. I cradled it tenderly, weeping with joy.

I did not know it then, but that vision of carrying the world in my heart would become one of the core images to which I would return in prayer in the decades ahead, a place of consolation that renewed my strength for climate activism. Years later, someone gave me a contemporary icon of Christ bending over the world, his arms embracing the planet.[1] I caught my breath in recognition. *Yes, that's right. That's just how it is.*

### 3. Climate Change and Addiction

Two years after starting my recovery I finished what I was doing, made a swerve, and headed to seminary. I needed to know: Who is the God who just saved my life? I was ordained in the Episcopal Church in June 1988. Not two weeks later, I picked up the New York Times and was startled by its front-page headline, "Global warming has begun (Shabecoff 1988)". NASA climate scientist James Hanson had testified to a congressional committee that scientists were becoming alarmed about the so-called "greenhouse effect" of burning fossil fuels. Human activity—driven by an economy dependent on coal, gas, and oil—was pushing the planet past its limits. The relentless extraction and burning of fossil fuels was polluting the global atmosphere with heat-trapping gasses; therefore, the atmosphere was rapidly heating. Scientists were concerned that the relentless consumption of dirty fossil fuels would disrupt the fragile balance of life. Great suffering lay ahead if we did not change course. We needed to stop what we were doing.

From that day forward, I began to track news about climate change. It became increasingly clear that the society in which I lived was behaving with the reckless abandon of an addict. In the ruthless push to drill oil wells, construct pipelines, blow off mountaintops, devour forests, and gobble up every last resource of the planet, we are laying waste to the land, air, and water upon which all life depends. The most vulnerable groups—low-income and Black, Brown, Indigenous, and people of color communities—are those hurt first and hardest by the effects of climate change, although even wealthy and privileged communities are beginning to suffer (Sengupta 2021). The resonance with addiction is haunting: as a society and a species we are caught up in highly destructive patterns of over-consumption and we have been unwilling or unable to quit.

In the months after James Hansen's testimony, a question emerged that became the riddle of my life, a question that fuels my vocation as a faith-based climate activist to this day: If God can empower a crazy addict such as me to make peace with their body, is it not possible that God can empower a crazed, addicted humanity to make peace with each other and the body of Earth?

### 4. The Shock of Climate Change

When I step outside this morning, I smell smoke. Haze blurs the heated air. Plumes of wildfire smoke that traveled thousands of miles across the country have reached us here in New England. With every breath, we inhale the residue of forests burning in western North America. Traces of distant trees that were set ablaze in massive fires sparked by unprecedented drought and heat now line our lungs. We are all connected.

Midway through the tumultuous, scorching summer of 2021, the damage caused by climate change is increasingly visible. Each day brings new reports of extreme heat, drought, fire, and floods. (Extreme precipitation is linked to global warming, because warmer air holds more water and therefore deposits more water when it rains—just as a larger bucket can hold and deposit more water). The American West and Southwest

are gripped by megadrought, an extraordinarily brutal and persistent drought which is draining reservoirs, withering fields, and increasing the spread of enormous wildfires. The Pacific Northwest, a usually cool and foggy part of the world, has roasted in record-setting levels of heat. Hundreds of people died in what one expert called "the most anomalous heat event ever observed on Earth".[2] North America is not the only place experiencing record temperatures—so, too, are the Middle East, South Asia, and Russia (Tharoor 2021). Meanwhile, torrential rains have drenched the mid-Atlantic. As much as ten inches of rain fell in southeastern Pennsylvania in under four hours. In China, terrified commuters riding subways stood on seats and clung to poles to avoid floodwaters from record-breaking rains.[3] Flooding recently killed hundreds of people in Central Europe, Uganda, Nigeria, and Italy. Famine stalks Madagascar as a drought tied to climate change dries up waterholes and crops. In Siberia, tens of thousands of square miles of forest are on fire, potentially releasing carbon into the atmosphere from the frozen ground below.

Today's headlines are frightening and stark, and they come in rapid succession. Fossil fuel emissions have disrupted Earth's atmosphere and biosphere even more quickly and dramatically than scientists predicted only a few years ago. If society is an addict dependent on coal, gas, and oil, then the addiction has reached its crisis point: Will we change course or will billions of us die, taking down with us the lives of countless other beings?

In a State of the Union address delivered in 2006, President George W. Bush warned of America's addiction to oil (Bush 2006). Of course, our dangerous relationship with fossil fuels does not function exactly like a substance addiction—we are not busily injecting oil into our veins in an effort to get high or experiencing DTs if access to coal is withdrawn. However, our society and economy—indeed, our whole way of life—does function like a person with a behavioral or process addiction: we are wretchedly, tragically—as a Christian, I would add "sinfully"—continuing to carry out activities that quickly or slowly will kill us and that are already killing countless people and other living beings worldwide. More than one Secretary General of the United Nations has called our present course "suicidal". Another word that comes to mind is "ecocidal". Indeed, a global panel of experts is now drafting a law to make ecocide—widespread destruction of the environment—a crime that can be prosecuted under international law (Saddique 2021; Surma et al. 2021).

## 5. Denial and Truth-Telling

What insights from the dynamics of addiction and recovery might inform our efforts to save what is left of the web of life and our struggle to preserve a habitable world? Six themes rise to the top: denial and truth-telling; isolation and community; grieving our losses; taking moral responsibility; praying the Serenity Prayer; and urgency, fear, and love.

Let us begin with denial and truth-telling. Built into addictive processes is the addict's insistent refusal or inability to perceive the reality or magnitude of the harm their behavior is causing themselves or others. Denial and minimization are characteristic ways that addicts avoid confronting their problem. As we wrote in *Rooted and Rising: Voices of Courage in a Time of Climate Crisis*, when it comes to facing the truth of climate change (Schade and Bullitt-Jonas 2019, pp. xx–xxi):

> The American public's widespread denial of climate change has had a stunning run. This is understandable, given that most people want to avoid thinking about something as deeply troubling as the Earth's climate crisis spinning out of control. We humans seem to have a built-in knack for delaying as long as possible the recognition of particularly troublesome facts. Some of us even turn denial and avoidance into a fine art. As comedian George Carlin observed, "I don't believe there's any problem in this country, no matter how tough it is, that Americans, when they roll up their sleeves, can't completely ignore".

However, we cannot ascribe the robust denial of climate change among many Americans solely to a supposed national capacity for dodging reality as long as possible. Nor should we assume that the denial of climate change and addiction to oil is a purely internal, mental problem that springs from a disorder in the brain, as one science writer has

proposed (Stover 2014). Nor is denial just a "defect of character", to use the language of the Twelve-Step Program—it is actually being generated and amplified by external forces, vested interests that have been hard at work since the late 1980s, spending billions of dollars in a deliberate campaign of disinformation to keep the American public confused about the reality, causes, and urgency of climate change (Oreskes and Conway 2011; Gelbspan 1997; Union of Concerned Scientists 2007).

Today, as Michael E. Mann explains in his masterful new book, *The New Climate War*, because the devastating impacts of climate change are now obvious in the daily news cycle, "the forces of denial and delay . . . can no longer insist, with a straight face, that nothing is happening. Outright denial of the physical evidence of climate change simply isn't credible anymore". As a result, fossil fuel corporations and oil-funded governments that continue to profit from our dependence on fossil fuels are shifting tactics to "a softer form of denialism" based on deception, distraction, and delay (Mann 2021, p. 3). This is what Mann calls "the new climate war", and the planet is losing.

Breaking through denial, whether its source be internal or external, is an essential aspect of climate activism. Climate activism faces outward: we have urgent work to do on the streets, in boardrooms, and in the backrooms where decisions are made. Mobilizing an effective, systemic response to the crisis at hand requires contending with political and corporate powers that seek to mire us in denial, distraction, and delay.

However, climate activism faces inward, too, as we reckon with our own layers of denial. You do not need to be a full-fledged climate sceptic who challenges the conclusions of mainstream science to be a person who slips into denial. Kari Marie Norgaard, a Professor of Sociology and Environmental Studies at the University of Oregon, has written helpfully about what she calls "the everyday denial of climate change, (Norgaard 2012)" the way that ordinary people who feel overwhelmed by the climate crisis simply change the subject to more manageable topics rather than face their guilt, fear, and helplessness. She connects this with the work of Robert Jay Lifton and Richard Falk, who studied, in relation to nuclear peril, "the absurdity of the double life": the way that people can live in two realities, being aware, on the one hand, of an enormous existential threat, while desperately clinging, on the other hand, to a pretense of conventional, ordinary reality.

We probably experience this cognitive dissonance in our own lives: although some part of us is aware that climate change looms over everything, we do our best to avoid thinking about it and we keep our focus on the immediate concerns of daily life. Friends of mine confess that even though they know that climate change is real, they do not pay very much attention to it: it is too painful to consider; they prefer to focus on more immediate, manageable concerns. In her brilliant novel, *Weather*, Jenny Offill evokes the difficulty of holding in mind both the close-in immediacy of our intimate, daily lives and the terrifying, large-scale reality of the unfolding climate catastrophe (Offill 2020).

Nevertheless, overcoming personal and collective denial is foundational to the ongoing work of recovering from addiction and creating a more just and sustainable future. As a recovering addict, I know how hard it can be to face, and keep facing, the truth: I remember how, in the early months of recovery, I needed to be reminded multiple times a day that I was a compulsive overeater and that a good day was a day in which I did not hurt myself with food. Unless I stayed in touch with allies in the Twelve-Step Program and unless I used its tools and carried out its Steps, it was simply too easy to slide back into denial and into the "stinking thinking" that led to relapse.

Similarly, as a faith-based climate activist, I must renew my commitment every day to dissolve my denial and to face reality as it is, not as I wish it were. That is not easy. As T.S. Eliot put it, "Humankind cannot bear very much reality (Eliot 1971, p. 118)". Can I make daily space in my mind and heart for the reality of climate change? Can I do something each day to keep myself informed, honor my emotional response, and carry out whatever actions I can that will contribute to healing? Just as an addict must renew her commitment to her own recovery daily, can we who live in an addictive society renew our commitment

to overcome denial of the climate crisis daily, and take some action, large or small, that leads to healing?

## 6. Isolation and Community

The Twelve-Step recovery process is carried out in community. Part of the power of the Twelve-Step model is the candor of its small group sharing: in every meeting, addicts seeking recovery share the truth of their lives and their desire to be sober (or drug-free or abstinent). We encounter each other as equals, because everyone, whether newcomer or old-timer, is in some sense a beginner and as dependent as anyone else on a power beyond themselves. In that circle of sometimes raw self-disclosure, we share our vulnerabilities and our experience, strength, and hope. Addiction is often called a disease of isolation, and by attending meetings, making phone calls, sponsoring and being sponsored, and carrying out acts of service, we gradually learn to find our place in a larger community. If, as Ann and Barry Ulanov so aptly put it, "Sin is the refusal to get our feet wet in the ocean of God's connectedness (Ulanov and Ulanov 1982, p. 96)", then the Twelve-Step model of healing in community is a release from sin. We are pulled into a current of connectedness that empowers us to set each other free: I may not be able to stop myself from overeating, but you can help me to stop; you may not be able to stop yourself from overeating, but I can help you to stop. To an addict who has white-knuckled countless lonely, failed attempts to kick the habit, entering the stream of relationships in a Twelve-Step Program can offer what feels like a miracle: buoyed by the support we feel all around us, it becomes much less difficult—perhaps even easy—to stay sober or abstinent, one day at a time. The antidote to addiction is connection.

I have never experienced a Twelve-Step meeting organized around recovery from addiction to fossil fuels or to exploiting the Earth,[4] but I understand the power of relationships to sustain my work as a climate activist. Who are the people to whom I can confess my confusion, fear, grief and outrage about the devastation of Earth and Earth's communities, both human and other-than-human? Who are the people seeking to move through their own despair and into a life of service? Who are the people trying to amend their lives so that they live more gently on the Earth and who inspire me to do the same? Who are the people committed to making sacrifices and taking risks for the sake of keeping fossil fuels in the ground and protecting life as it has evolved on this planet? These are some of the people I want to be close to, because I can learn from them and grow with them. Even if we never sit together in one room, even if they live someplace far away—indeed, even if I never meet them and never even learn their names—they are my circle of support, allies in my own struggle to live in harmony and balance with Earth.

"Don't talk, don't trust, don't feel"—those three core rules of alcoholic and dysfunctional family systems were laid out by Dr. Claudia Black years ago in her seminal book, *"It Will Never Happen to Me!"* (Black 1981). Some of the other rules include "don't think" (about what is going on) and "don't question" (what is happening). Whenever we gather to talk honestly about the climate crisis, trust each other with our truth, dare to feel our feelings, think about what is going on, and ask questions about what is happening, we transgress those dysfunctional dynamics and begin to build a more authentic and resilient network of relationships. Simply breaking the silence around climate change—speaking honestly to a friend about one's worry or concern—can be the beginning of release from the paralyzing isolation that tells us that climate change is too big, too frightening, or too political to discuss.

Experiencing the healing power of connections extends to our relationship with the natural world. Just as addicts generally treat their bodies with violence or contempt, so most of us in today's dominant culture were raised to override and ignore the needs of the living world around us. Nature was supposed to be at our beck and call, a limitless resource that human beings were entitled to drain—nothing more than commodities to be bought, sold, processed, consumed, and discarded. Many Westerners are only beginning to acknowledge our deep alienation from the rest of the created order and are only now

discovering the deep wisdom of Indigenous traditions and our own mystical traditions, which speak of the essential interconnectedness, sacredness, and mutuality of everything that exists.

Learning to cultivate loving, life-giving relationships with other people and with the other creatures and elements with whom we share the planet is medicine for addiction of every kind.

### 7. Grieving Our Losses

Facing addiction requires facing grief. Addicts who are beginning their journey of recovery will likely have many losses to grieve, such as a failed marriage, a lost job, a damaged reputation, or estranged co-workers, children, and friends. Furthermore, in relinquishing their drug of choice, addicts are also losing what seemed to be their lover or best friend, the substance or behavior to which they clung—even if they hated it—in order to manage their life. Not only that, when addicts stop using their drug, the feelings that had been suppressed by their compulsive behavior will likely come surging back into awareness: grief, shame, fear, anger, loneliness, confusion, the whole nine yards. Living into recovery, a day at a time, can be an emotionally turbulent process.

Confronting the climate crisis likewise requires acknowledging grief and other painful feelings. Grief is the normal, healthy response to loss, but the dominant culture in which we live does not handle grief well. Many of us tend to sidestep or suppress our grief, fearing that we will look weak, sentimental, morbid, or pathetic. We may also avoid thinking about climate change because we fear being overwhelmed by our emotions. What can we possibly feel in response to the acidifying ocean, the children choking from asthma in our inner cities, the rising seas, the ever-increasing droughts and floods, and the cascade of species being made extinct? Who wants to allow an emotional response to hearing that climate change is already making parts of the world too hot and humid for humans to survive (Mellen and Neff 2021)? Or that unchecked climate change could collapse whole eco-systems quite abruptly, starting within the next ten years (Berwyn 2020)? Or that the natural world is at a far greater risk from climate breakdown than was previously thought (Harvey 2020)? Stunned by the gravity of news such as this, many of us feel helpless and turn away. The scale of the problem feels too big in comparison with our one small life and our limited powers. We might as well cling to business as usual for as long as we can—drive, shop, send the kids to school, earn the promotion, fix supper, check social media—and let someone else handle the bigger problem, maybe the experts or maybe future generations. We might as well stay distracted, busy, and numb. We might as well zone out for as long as possible.

Emotional withdrawal is a natural response to trauma. We are all living in the context of ongoing and accelerating global trauma, even if our corner of the world has not yet borne the full brunt of climate change. It is understandable if we are inclined to anesthetize ourselves and shut down emotionally. However, shutting down is its own form of suffering. As Franz Kafka observed, "You can hold yourself back from the sufferings of the world, that is something you are free to do and it accords with your nature, but perhaps this very holding back is the one suffering you could avoid".

It is easier to release into grief when we feel supported, understood, and upheld. This is where the power of community comes in. Like addicts recovering in the Twelve-Step Program, we do not have to tremble in fear or shed tears alone. A variety of circles have formed in recent years to help participants grapple with the spiritual and existential questions raised by climate emergency and other forms of collective trauma. Among others, they include The Work That Reconnects, based on the teachings of Joanna Macy; Rabbi Jennie Rosen's organization, Dayenu; and Margaret Klein Salomon's Climate Awakening.[5] Psychological and psychiatric associations are increasingly aware of the mental health challenges posed by social and ecological breakdown and are training clinicians to address these issues in their work with clients.[6] Parish leaders also have a golden opportunity to gather members of their congregation for prayerful, small-group conversations about

climate change and to create communities of truth-telling that allow the honest expression of pain.

We are blessed that many faith traditions provide tools and rituals for accessing and processing grief. Learning practices of contemplative prayer and meditation can be helpful, because they give traumatized people a technique to calm down, steady the mind, and quiet the nervous system. Contemplative prayer, often defined as "a long, loving look at the real", resonates with the Zen teaching, "Stay present to what's happening". In a time of emotional turbulence and agitation, contemplative prayer can help us cultivate trust and patience. We learn to sit still in the midst of uncertainty, to wait in the darkness, to relinquish our anxious and futile quest to stay in control, and to listen for the inner voice of love. To cite the psalmist: "Be still . . . and know that I am God" (Psalm 46:11).

From out of the stillness, feelings arise that may need expression—even visceral, bodily expressions, such as wailing, stamping, dancing,[7] drumming, and singing. Expressive prayer is essential to articulating grief, whether we do it together or alone. Lament is an ancient form of prayer found in the Psalms, in the prophets, and in the words and actions of Jesus. He wept at the death of Lazarus, he wept over the city of Jerusalem, and he cried out to God on the cross, using the lament of Psalm 22. Lament is not self-pity nor is it simply whining. Lament is a deep outpouring of sorrow to God. Learning how to pray with painful feelings can help us to grow in intimacy with God and to experience solidarity with everyone who suffers (Bullitt-Jonas 2000). Spiritual directors with an awareness of the dynamics of addiction can help the people they guide to explore pathways of prayer that allow the expression of feelings (Bullitt-Jonas 1991).

Lament, especially public lament, can be empowering. Theologians such as Walter Brueggemann (Brueggemann 1978; Sharp 2011, pp. 179–205), drawing on the work of Dorothee Soelle, Jurgen Moltmann, and Abraham Heschel, have brilliantly shown us that lament is the beginning of criticism of an unjust social order. Articulating anguish and experiencing passion—defined as "the capacity and readiness to care, to suffer, to die, and to feel (Brueggemann 1978, p. 41)"—is the enemy of any society built on ignoring the cries of the marginalized and oppressed, the cry of the Earth and the cry of the poor. Lament can end in hope or praise, because in lament we experience the presence of a living, loving, and liberating God. Lament can lead to action, because the more we experience our unshakable union with a love which is stronger than death, the freer we will be to take actions commensurate with the emergency in which we find ourselves.

The climate crisis brings us to our knees. It also brings us to our feet.

## 8. Taking Moral Responsibility

Basic to the process of recovery in the Twelve-Step Program is taking moral responsibility for one's actions. Addiction is not "a moral issue", if by that we mean that addicts are "weak" or "bad" people without moral principles; in fact, addicts are people with a complex medical disease or condition. However, addiction does have a moral dimension: you cannot be set free from addictive behavior unless you carry out a deep housecleaning. Seven of the Twelve Steps (Steps 4–10) engage recovering addicts in a thorough and ongoing process of growth in moral self-awareness, accountability, and responsibility.

Reckoning with our moral responsibility for contributing to the climate crisis is complex (Jenkins 2008, 2013; Moore and Nelson 2010; Northcott 2007; Rasmussen 1996). Climate change is a justice issue on many levels. For starters, it is an issue of *social and economic* justice, because impoverished individuals, communities, and nations are those who suffer the effects of climate change first and hardest; they are the ones least able to adapt, and the ones least likely to have a seat at the table where policy decisions are made. Climate change is also an issue of *international* justice. As the Union of Concerned Scientists points out, "The world's countries emit vastly different levels of heat-trapping gases into the atmosphere (Union of Concerned Scientists 2008)". Climate change is caused mostly by the wealthy nations—developed countries and major emerging economies lead in total carbon dioxide emissions—but it is the poorer nations which are most vulnerable to its painful

effects. The question of international justice becomes even more pointed when considering the per capita consumption of fossil fuels. Saudi Arabia and the United States are tied in first place for the world's highest per capita carbon emissions, far outpacing the per capita outputs of poor nations (Statista 2021). One analysis reviewed public health studies of the effects of burning fossil fuels and concluded that the lifestyles of about three average Americans create enough planet-heating emissions to kill one person (Millman 2021).

Climate change is a matter of *intergenerational* justice, because right now we are stealing a habitable Earth from our children and our children's children. If we continue with business as usual, we will leave a ruined world to those who come after us. No wonder so many members of the Sunrise Movement [8] and so many other young climate activists are angry!

Climate justice is likewise inextricably linked to *racial* justice. In the piercing words of Hop Hopkins, the Sierra Club's Director of Organizational Transformation, "You can't have climate change without sacrifice zones, and you can't have sacrifice zones without disposable people, and you can't have disposable people without racism (Hopkins 2020)".

Perhaps we must speak of *interspecies* justice, as well, because for the first time in the planet's history, a single species, *Homo sapiens,* is in the process of wiping out vast populations of other creatures, and even entire species. Driven by climate change and other pressures of human activities, the world's wildlife populations have plummeted by more than two-thirds in the last 50 years, according to a 2020 report by the World Wildlife Fund (Rott 2020). We are also in the midst of Earth's sixth extinction event. With dismay, scientists are describing what they call a "biological annihilation (Ceballos et al. 2017)". Recognizing that we are now in an emergency that threatens human civilization, one expert commented, "This is far more than just being about losing the wonders of nature, desperately sad though that is . . . This is actually now jeopardizing the future of people. Nature is not a 'nice to have'—it is our life-support system (Carrington 2018)".

To push away the horror—and the responsibility—it might be tempting to shift the blame for the climate crisis onto the generations that preceded us. "After all", we may tell ourselves, "burning fossil fuels began long before I was born; people have been burning fossil fuels since the eighteenth century, when the Industrial Revolution began". However, adults such as me cannot get away with that attempt at moral deflection (which is so characteristic of an addict): more than half of all $CO_2$ emissions since 1751 were emitted in the last 30 years (Stainforth 2020). That is, in a single lifetime—ours.

Clearly, the climate crisis is not only a scientific, political, economic, or technical issue—it is a moral issue, as well. What if members of a high-carbon, high-consumption society faced our guilt and took Step 4 ("Made a searching and moral inventory of ourselves")? What if we carried out the Steps that follow and took bold, even radical action to address the moral injustice of climate change?

Taking personal responsibility means that each of us does our part to solve the problem. Many of us start reducing our personal and household "carbon footprint". We recycle, we buy less stuff, we eat less meat and move toward a plant-based diet. We do whatever we can afford to do—install solar panels, buy an electric car, eat local, organic foods, upgrade insulation, turn down the heat, use less air conditioning. Taking these kinds of personal steps to reduce our carbon footprint is worthwhile in many ways: they align our lives more closely with our values; they can inspire friends and neighbors to follow suit, making it socially acceptable and morally normative to live more gently on Earth; and they relieve our sense of cognitive dissonance—we know that we are taking action to address an existential crisis. After all, as Lao Tzu said, "A journey of a thousand miles begins with a single step". Making personal changes in lifestyle may be that vital first step on the ramp to more effective action.

However, do not be fooled—if we limit taking personal responsibility simply to changing our lifestyle and consumer choices, we are falling for the lie that individual behavior is enough. It is not. Turning off the lights and driving an electric car may be the right thing to do and make us feel morally "cleaner", but moral action only makes a

substantive difference when we join the fight for systemic change. A societal transformation from top to bottom is what is required to avert climate chaos—that is what the world's pre-eminent climate scientists told us in the 2018 report from the U.N.'s Intergovernmental Panel on Climate Change. The only way to do that is to push for collective solutions, to become politically engaged, and to make it politically possible to do what is scientifically necessary to maintain a habitable world.

In the meantime, fossil fuel corporations are working hard to shift responsibility for the damage that their products cause (damage that these companies concealed and denied for decades) to individual consumers. Like drug dealers, they make a fortune by pushing a deadly product and then blame their customers if they buy it and become sick. A fascinating article by Amy Westervelt explains how, for over 100 years, various industries, including tobacco, beverage packaging, guns, and fossil fuels, "have weaponized American individualism, laying the blame for systemic issues at the feet of individual citizens".[9] Westervelt observes that BP "famously invented the ultimate tool for pinning greenhouse gas emissions on individual consumers: the carbon footprint calculator".[10] As she points out:

> This rhetorical framing flourishes not only because it taps into America's individualistic identity, but also because it presents easy solutions: simply buy different things in your own life, walk or bike a bit more, and everything will be fine! It also provides a purity test that no climate activist can possibly pass. It's the perfect setup for oil companies: The problem is consumers, not industry, and no consumer can ever reduce their carbon footprint enough to be a credible critic. (Westervelt 2021)

Framing the climate crisis in moral terms gives us an opportunity to understand that effective moral action includes collective moral action. To be blunt, do not be a consumer, be a citizen.

The scope and speed of the climate crisis require more than personal changes in behavior—they require collective action and a push for policies such as pricing or regulating carbon, eliminating fossil fuels subsidies, providing incentives for clean renewable energy, and ensuring that historically marginalized communities enjoy the benefits of clean energy.

Climate scientists are increasingly concerned that if global warming continues unchecked, the Earth will soon pass so-called "tipping points" beyond which possibly irrevocable disaster will ensue (Harvey and Agencies 2021). Is it possible to create a *social* tipping point that would propel a swift transition to clean energy? According to one study (Otto et al. 2020), providing a moral framework for the climate crisis would contribute to a social tipping point and help activate "contagious and fast-spreading processes" that lead to global decarbonization. Using a term from the field of addiction, the study argues that revealing the moral implications of fossil fuels is an "intervention" that would accelerate a rapid global transformation to carbon-neutral societies. Let us start this addict on the road to recovery.

## 9. Praying the Serenity Prayer

Like most recovering addicts in the Twelve-Step Program, I frequently turn to the Serenity Prayer: "God, grant me the serenity to accept the things I cannot change, courage to change the things I can, and wisdom to know the difference". Based on a longer prayer by theologian Reinhold Niebuhr, these words have helped countless addicts to search their minds and hearts as they sort out what to hold on to and what to let go, what is theirs to do and what is not. Implicitly, the prayer invites us to rein in our compulsive craving for control and to find peace even in the midst of trouble. It rouses us from passivity and inertia so that we change what we can (and should) change. Additionally, it recognizes that we do not see these things clearly, and need to ask for God's help.

The prayer is immensely useful for everyone concerned about climate change. What is it that I need serenity to accept? What is it that I need courage to change? How do I know which is which? The questions themselves drive me into prayer, and the answers change over time as I listen and learn. I pray for serenity to accept the reality of the climate

crisis and the painful manifestations of that crisis which emerge every day—and I find my way to serenity only as I pray my way through outrage, fear, and grief. I pray for courage to change the things I can—and I find that courage only as I keep entrusting my actions to God. I pray for the wisdom to know what is and is not mine to do—and I try to forgive myself when I get that wrong. The Serenity Prayer is pithy, enigmatic, and as pure as prayer comes—it does not give answers; it simply opens a door to God.

We bring into prayer what we know about the world, so it is good to be aware that many internal and external forces are at work, insisting that there is little we can do to slow climate change. I will mention only two. One is external: fossil fuel corporations are eager to amplify our supposed helplessness to quit using their products. They are delighted when "collapse-aware" people throw in the towel and accept that we are doomed, that it's too late to take effective active to stave off climate catastrophe. As Michael Mann explains, "Doomism potentially leads us down the same path of inaction as outright denial of the threat". He adds, "The surest path to catastrophic climate change is the false belief that it's too late to act (Mann 2021, pp. 179, 223)".

A second message that dampens courageous action is internal: without knowing it, we tend to accept an increasingly degraded natural world as normal. It has been called "shifting baseline syndrome" or "sliding baseline syndrome": each generation adapts to worsening circumstances over time, disregarding the abundance that previous generations knew, while peacefully accepting what remains as fine, or to be expected. We slowly adjust to unthinkable circumstances. As David Roberts explains, the scariest thing about global warming is that we could grow accustomed to it—grow used to massive fires, severe flooding, killing levels of heat—and never experience a moment of reckoning. We could sleepwalk our way to catastrophe (Roberts 2020; Campbell 2020).

Humans have been a successful species partly because we are so adaptable, but the capacity to adapt can also be a moral and even mortal liability. I think of the bitter comment uttered by Raskolnikov, the anti-hero of Dostoevsky's *Crime and Punishment*: "Men are scoundrels; they can get used to anything (Dostoevsky 1989, p. 22)!" I also think of the less bitter, but still bracing quote attributed to Thomas Merton: "The biggest human temptation is to settle for too little".

When does our purported serenity to accept the things we cannot change in fact mask our apathy and amnesia? When does serenity camouflage the refusal to care—what Fr. James Keenan calls "the failure to bother to love"? Rabbi Abraham Heschel insisted that "Prayer is meaningless unless it is subversive, unless it seeks to overthrow and to ruin the pyramids of callousness, hatred, opportunism, falsehoods". Subversive prayer breaks through cheap serenity. True serenity springs not from choosing comfort and avoiding conflict, but from the desire to seek only God's will, to abide in God's love, and to carry out what love requires, even when doing so is costly or difficult.

Once upon a time in the United States, people accepted many things as normal—slavery, Jim Crow, child labor, 80-h work weeks, the disenfranchisement of women and African Americans, the indiscriminate use of DDT, and so much more. What awoke them from their "serenity" was the persistent, massive, collective efforts of countless ardent people who were unwilling to settle for so little. What is it that we, too, must refuse to accept as normal? Are we willing to join the movements now rising up around the world—the climate justice movement, the human rights movement, the Indigenous rights movement, and the coalitions—both faith-based and secular—that are pressing to eliminate dirty emissions, restore a safe climate, reverse the sixth mass extinction of species, and create a just society that works for everyone?[11]

## 10. Urgency, Fear, and Love

People suffering with addiction do not walk casually into a Twelve-Step meeting. We are not there to pass the time. We are not there to virtue signal. We are not there to pass a purity test. We are there to save our lives. Urgency is what drives a person into recovery. We have reached the point of admitting, as the Big Book of Alcoholics Anonymous puts

it, that "half-measures availed us nothing"[12]—not launching another diet, not drinking only on weekends, not shooting up just once in a while. We need a thorough makeover, a transformation which is physical, emotional, and spiritual.

Urgency is what today's climate prophets are conveying. Scientists speak with alarm about the very short time we have left in which to safeguard a stable climate; they speak about the urgent need for "rapid and far-reaching (United Nations Sustainable Development Goals 2018)". changes in all aspects of society. We cannot miss the urgency of Greta Thunberg, the Swedish teenager with the round face, straight blonde hair, and fierce, unyielding eyes, who spoke with such intensity to the U.S. Congress, the U.N. COP meeting, and the World Economic Forum, telling the world, telling the adults who failed to take action: "The house is on fire". Our planetary home is on fire. It is going up in flames.

It is a precious moment when an addict listens, grasps the urgency, feels the heat, and makes the decision to choose life. It is a precious moment when an addict admits that their life is unmanageable, that they need help beyond themselves, and that the time has come for decisive action. It is a precious moment when an addict realizes that the old way of life has to die in order for new life to be born. Will our generation be able to look back with gratitude one day and sing "Amazing Grace"?

Fear is what forced me into recovery, and fear may be what forces society to awaken to the climate crisis at last. Given the predicament in which we find ourselves, we have good reason to be afraid. However, fear cannot sustain us over the long haul—only love can do that.

Therefore, I thank God for all the people who are willing to face their fear, to empathize with other people's fear, and to stand together. I thank God for all the people who refuse to turn away from each other or against each other, but who decide instead to turn toward each other, to join forces and join hands. I thank God for the deep message of all the world's religions: we are interconnected with each other and with the web of life.

As an addictive society wakes from its restless, deathly sleep, faith communities can help to restore our capacity to love God and neighbors. In a sermon, D'var Torah, and dharma talk; in prayer groups, worship services, and meditation groups; in pastoral care, outreach, and bold public advocacy, communities of faith and spiritual practice can renew our intention and deepen our capacity to act in loving ways, to respect the dignity of every human being, and to cherish the sacredness of the natural world. Faith communities speak to the heart of what it means to be human. When people are closing their eyes to a crisis or going mad with hatred and fear, only love can restore us to sanity.

We can be more than addicts on a self-destructive path. Additionally, we can be more than chaplains at the deathbed of a dying order. We can be midwives to the new and beautiful world that is longing to be born.

**Funding:** This research received no external funding.

**Institutional Review Board Statement:** Not applicable.

**Informed Consent Statement:** Not applicable.

**Conflicts of Interest:** The author declares no conflict of interest.

## Notes

[1] Robert Lentz, OFM, "Compassion Mandala", https://robertlentzartwork.wordpress.com/2012/06/19/httpswww-trinitystores-comstoreart-productsrlcmm/ (accessed on 27 July 2021).

[2] Christopher Burt, quoted by (Cappuci 2021).

[3] This example and those that follow are cited by (Kaplan and Dennis 2021).

[4] One very interesting initiative that weaves together addiction/recovery, Christian faith, and care for the Earth is EcoFaith Recovery. Based in the Pacific Northwest, EcoFaith Recovery is "a leadership development effort grounded in the Christian tradition and welcoming all who seek recovery from societal addictions to unsustainable ways of life. Our recovery begins as we come out of isolation and rediscover our relatedness to God, ourselves, each other, and the entire earth community of which we are a part". See: http://www.ecofaithrecovery.org/ (accessed on 31 August 2021).

5     https://workthatreconnects.org/, https://dayenu.org/, https://climateawakening.org/ (accessed on 30 July 2021).

6     See, for example, Climate Psychology Alliance https://www.climatepsychologyalliance.org/ (accessed on 31 August 2021) Climate Psychology Alliance North America https://www.climatepsychology.us/ (accessed on 31 August 2021) and Climate Psychiatry Alliance https://www.climatepsychiatry.org/ (accessed on 31 August 2021).

7     In 1992, Joanna Macy brought the Elm Dance to people living in areas that had been poisoned by the Chernobyl disaster. This simple circle dance, now associated with The Work That Reconnect, is intended for all who experience collective trauma, https://workthatreconnects.org/resources/elm-dance/ (accessed on 31 July 2021).

8     The Sunrise Movement is a youth movement to stop climate change and create millions of good jobs in the process, https://www.sunrisemovement.org/ (accessed on 31 August 2021).

9     (Westervelt 2021). In *The New Climate War*, Michael E. Mann addresses this topic in a chapter entitled, "It's YOUR Fault", pp. 63–97.

10    "Calculate and offset your emissions", https://www.bp.com/en_gb/target-neutral/home/calculate-and-offset-your-emissions.html (accessed on 31 August 2021).

11    See, for instance, The Climate Mobilization, Indigenous Environmental Network, 350.org, Poor People's Campaign, Sunrise Movement, Extinction Rebellion, Mothers Out Front, Interfaith Power & Light, GreenFaith, The Shalom Center, Dayenu, and many others.

12    https://www.aa.org/assets/en_us/en_bigbook_chapt5.pdf (accessed on 31 August 2021).

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
