# Peer review of "Climate Change, Addiction, and Spiritual Liberation"

_religions, doi:10.3390/rel12090709_

Round 1

Reviewer 1 Report

Line 159 – Is melodramatic. It suggests probable human extinction caused by climate. Beyond the minds of alarmists such as Guy MacPherson in the USA, this is not based on well-founded science.

Lines 165 – 171 – This reads like heresay. Sinful is not defined, it is assumed that the reader will know what is meant. In what way ecocide might be legally defined is not stated (that is a problem with the late Polly Higgin’s approach generally). If this paper is to be considered academic, an overly chatty and ill-defined format is not appropriate.

Up to line 246 – much of this is in a chatty rather than academic mode. I don’t know what is being looked for in this edition, but if a scholarly approach is required, I would suggest cutting what has preceded by about half, and succinctly stating only the points that clearly set out the predicament. If, however, your journal aims for a more pastoral or popular approach, then the way this is written is OK as it invites the general reader in.

Line 257 – I take back what I said about sinfulness – s/he defines it here. And satisfactorily.

Lines 433 – 437 leaning on Wallace-Wells is academically uncomfortable. Some of his work has been criticised by climate scientists as alarmist and, certainly, journalistic. The Michael Mann book that’s in the biblio has 12 pages on this. I would suggest reading them, and reconsidering W-W.

Overall, this writer’s strength lies in their experience of addiction. They are not so strong on evaluating climate science beyond the pedestrian. I would suggest that the former is kept but the latter cut down to a minimum. It can be assumed that the intelligent reader will already know most of what the author has stated about climate change, but some of the spiritual insights around addiction will not be widely known. I would suggest that the orientation should be towards a pastoral readership.

Author Response

Thank you for your thoughtful comments on my essay.  As you surmised in several of your comments, I am aiming in this essay for the general reader, especially the reader who may be somewhat concerned about climate change but who has not yet reckoned emotionally and spiritually with the full-blown crisis.  Although this paper is well-researched, it is not intended to be a contribution to scholarly thinking but instead to be a more personal and pastoral essay that reaches readers on a gut level.  Addicts can’t think their way out of their dilemma – we need to be reached (converted, transformed) by an appeal to the heart as well as the head. I seek in this paper to evoke on an emotional level the true stakes of the predicament in which we find ourselves and to motivate action.  I agree with you that this article is aiming for a pastoral readership.

Line 159: Per your suggestion, I revised the sentence so that it makes no claim about human extinction. 

Lines 261-262: I am glad that this working definition of sin makes sense to you.   

Lines 438-439, 442: Thanks for the suggestion to go back to Michael Mann’s book regarding Wallace-Wells.  I do like the two Wallace-Wells sentences that I quote – they are so accurate and so succinctly stated! – but given the controversy about his work, I accepted your suggestion and dropped them.  I don’t want to distract the reader. 

I understand that you recommend cutting out most of the climate science, but I would like to retain it, such as the list of this summer’s climate events and the brief summaries of the situation.  I am trying not to evaluate climate science but rather to remind the well-read reader (or to inform the not-so-well-read reader) of the magnitude of the problem and the need for action. 

I hope you will find that the revised essay addresses your concerns.  Thanks again for your time and expertise.  

Reviewer 2 Report

The reflection is a thoughtful and well written. Using the steps within the AA process to address climate change is unique and provides an emotional connection that is powerful. The approach to climate change is something that needs to draw on the visceral experience. There is an issue. In lines 180 to 182, there seems to be something missing...either a whole sentence or paragraph. The end of the paragraph ends on line 181 with a comma. Then the next paragraph begins with what seems like another thought. This should be addressed after revision. 

Author Response

Thank you for taking the time to read and respond to my essay, and for your thoughtful comments.  I am glad that you found the essay emotionally powerful and that my effort to connect the process of recovering from addiction with the process of responding to the climate crisis resonated with you.  As for the apparently missing paragraph, something in the formatting was askew.  The quotation is there – it just got folded into the next section of text. Thanks again for your help and for your sympathetic reading of my article.  I made some revisions and I hope you will like them.  Thanks again for your attention and care.